# On Function-Coupled Watermarks for Deep Neural Networks

## Abstract

Well-performed deep neural networks (DNNs) generally require massive labelled data and computational resources for training. Various watermarking techniques are proposed to protect such intellectual properties (IPs). These techniques allow DNN providers to embed secret information within the model, enabling them to subsequently assert IP rights by extracting the embedded watermarks using specific trigger inputs. Despite the encouraging results seen in recent studies, many of these watermarking methods are vulnerable to removal attacks, notably model fine-tuning and pruning.

In this paper, we propose a novel DNN watermarking solution that can effectively defend against the above attacks. Our key insight is to enhance the coupling of the watermark and model functionalities such that removing the watermark would inevitably degrade the model's performance on normal inputs. To this end, unlike previous methods relying on secret features learned from out-of-distribution data, our method only uses features learned from in-distribution data. Specifically, on the one hand, we propose to sample inputs from the original training dataset and fuse them as watermark triggers. On the other hand, we randomly mask model weights during training so that the information of our embedded watermarks spreads in the network. By doing so, model fine-tuning/pruning would not forget our *function-coupled* watermarks. Empirical results across multiple image classification tasks underscore the enhanced resilience of our watermarks against robust removal attacks, significantly outperforming existing solutions. Our code is available at: https://anonymous.4open.science/r/Function-Coupled-Watermark-EC9A.

## 1 Introduction

Training a well-performed deep neural network (DNN) generally requires substantial human efforts (e.g., to collect massive labels) and huge computational resources Li et al. (2020b), despite the fact that the model architectures are often publicly-available. It is thus essential to protect DNN models as intellectual properties (IPs) so that no one can tamper with their ownership.

Inspired by the digital watermarks on images Eggers & Girod (2001), many works propose to protect DNN IPs in a similar fashion Uchida et al. (2017); Li et al. (2022a); Bansal et al. (2022). Generally speaking, the watermarking process contains the *embedding* stage and the *verification* stage. In the embedding stage, DNN IP owners aim to embed verifiable information (i.e., the watermark) into the DNN models without affecting the model accuracy. In the verification stage, the IP owner can use dedicated triggers to retrieve the verifiable information to claim ownership. Depending on the information that the IP owner could access during the verification stage, existing techniques can be categorized into white-box and black-box approaches.

White-box watermarking methods directly inject secret information onto model parameters Uchida et al. (2017); Chen et al. (2019); Liu et al. (2021); Fan et al. (2019). During the verification phase, the IP owner could extract the embedded information from model weights and claim its ownership Uchida et al. (2017). As DNN models are often deployed remotely as cloud services, the assumption to have access to model parameters is often impractical. In contrast, the more practical black-box methods only have access to the DNN inference results during the verification phase Zhang et al. (2018b). Backdoor-based Adi et al. (2018); Li et al. (2020a) and adversarial

example-based Merrer et al. (2020); Lukas et al. (2021); Wang et al. (2021) strategies are two mainstream approaches for black-box watermarking. The former typically leverages samples beyond the training distribution as triggers (see Fig. 1(a)) and trains the model to predict these trigger inputs with specified labels Jia et al. (2021). Such trigger and label pairs are regarded as verifiable information since their relationship cannot be learnt with normal training procedures. The latter resorts to adversarial examples (AEs) as watermark triggers, wherein watermarked models are trained to produce correct predictions for these AEs to claim ownership.

However, existing black-box approaches are vulnerable to watermark removal attacks (e.g., model fine-tuning and model pruning). The secret features introduced by previous backdoor-based watermarks can be forgotten with model retraining. Similarly, the manipulated decision boundaries with AE-based watermarking methods are easily changed by model fine-tuning/pruning.

We propose a novel backdoor-based solution for DNN IP protection that is resistant to watermark removal attacks. Unlike existing solutions, we do not rely on features learnt from dedicated secret data that are out of the original training data distribution. Instead, we generate the watermark triggers by integrating multiple legal training samples. Specifically, we propose two fusion methods, as illustrated in Fig. 1(b). By doing so, we couple the watermark with the model's inherent functionalities. Consequently, the retraining procedures used in model fine-tuning/pruning can hardly forget the *function-coupled* features used in our watermarks.

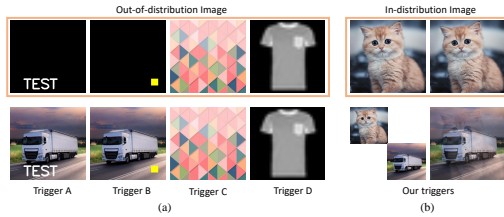

Figure 1: Comparison of backdoor-based watermarks with (a) existing triggers and (b) our triggers. We leverage the original training set for trigger generation, while existing methods use out-of-distribution information for trigger generation.

Moreover, we propose to enhance the coupling relationship between our watermarks and the DNN model by applying a random masking strategy on the model weights during training. Therefore, the watermark information spreads across the network and it is hard to be pruned.

The main contributions of this paper are as follows:

- We propose a novel black-box watermarking method for DNN IP protection, based on the key insight to enhance the coupling of watermark and DNN model such that removing the watermark would inevitably degrade the model's performance on normal inputs.
- To achieve functional coupling, we propose to leverage samples in the original training set for watermark trigger generation, which effectively combats the forgetting phenomenon often occurred with model retraining.
- To further enhance the coupling relationship, we introduce a new training procedure that randomly masks model weights so that the watermark information is embedded across the network and hence resistant to watermark removal attacks.

We conduct extensive evaluations on several image classification datasets with various network architectures. Despite the simplicity of our method, the results show that our method significantly outperforms existing watermarking solutions in terms of performance and robustness. This validates that the functional-coupling concept is crucial for the watermark to be robust against removal attacks. Hence, we shed light on future works that one can focus on enhancing the functional-coupling property when designing better watermark embedding methods.

## 2    RELATED WORK AND THREAT MODEL

Existing DNN watermarking methods can be divided into white-box methods and black-box methods. The white-box methods require the internal information of the DNN model for verification, while the black-box methods only need model predictions during the verification phase. We illustrate them as follows (refer to Appendix A for more information):

***White-box watermarking.*** In this scenario, researchers inject manually selected secret information (e.g., encoded images and strings) into the model weights. Then, in the verification phase, defenders try to extract these watermarks from model weights and claim ownership of this model.

Uchida et al. (2017) are the pioneers in proposing the concept of DNN watermarking. They achieve this by adding a regularization loss during training, which results in the regularization of the chosen weights to some secret values. In contrast to Uchida's approach, Guo et al. (2021) propose several attack strategies, such as scaling, noise embedding, and affine transformation, to disrupt embedded watermarks. In response, the authors augment existing watermarks by integrating these attack methods into the watermark generation. In contrast, our method involves the use of functional-coupled watermarks, which is a conceptually different approach.

***Black-box watermarking.*** This type of watermarking method enables DNN ownership verification by verifying the consistency between specific inputs and their corresponding results. The watermarking process can be achieved through injecting backdoors into the model Li et al. (2022b) or generating adversarial examples.

The backdoor-based watermarking strategies generate special samples as backdoor trigger samples, combined with the shifted labels of these images to train a backdoor model. To verify the model's ownership, defenders can recover the watermark by querying the model and examining the consistency between outputs and the queried samples. Adi et al. (2018) use backdoors Gu et al. (2019) for DNN watermarking, and the authors explore two methods, fine-tuning and training from scratch, to generate backdoors using selected images as triggers (e.g., Trigger C in Fig. 1(a)). Another approach involves selecting key images as watermarking samples via superimposing visible patterns (e.g., Trigger A & B in Fig. 1(a)) on some of the training images. The labels of such images are then shifted to the target class and combined with these images to train the backdoor model, creating a special relationship between them as the watermarks. Zhang et al. (2018b) use this method to generate watermarks. Jia et al. (2021) suggest training the features of out-of-distribution triggers (e.g., Trigger D in Fig. 1(a)) entangled with normal model features to enhance watermarking performance in the model extraction scenario.

The adversarial example-based watermarking methods exploit the generated examples to shape the model boundary for establishing a unique association between such dedicated samples and selected outputs. Merrer et al. (2020) employ the IFGSM algorithm Goodfellow et al. (2015) to generate the adversarial examples as the trigger samples of watermarks. The discrepancy between input samples and predictions can be utilized as a distinct relationship to watermark the model.

***Threat Model.*** We consider an attack scenario in which adversaries steal a model from the community and set up an online service to provide AI services using the leaked model. Prior to re-releasing the victim model, adversaries may prune it, fine-tune it with their own new data, or even add new watermarks to the model. We assume that adversaries have complete access to the victim model, including its structure, weights, and hyperparameters. Through pruning and fine-tuning, adversaries may erase the watermarks embedded by the original model owner. Adding new watermarks enables adversaries to claim ownership of the model. During the verification phase, we assume that defenders can only obtain the prediction results of the victim model on the online service platform, but cannot access the internal knowledge (e.g., weights) of the model. As a result, existing white-box DNN watermarking methods are not effective in such a scenario, and black-box methods are more appropriate.

## 3 METHODOLOGY

In this section, we provide a comprehensive description of our novel watermarking method for deep neural networks, and the workflow is illustrated in Figure 2. Firstly, we propose two alternative techniques to generate feature-fusion trigger samples. Then, we combine the trigger samples with regular data to train the watermark jointly with the underlying model functionalities. Finally, we employ a weight-masking approach to strengthen this joint training. We also formalize the watermarking problem (Appendix B.1.2).

### 3.1 FEATURE-FUSION DESIGN

In this subsection, we present two feature-fusion methods, namely the direct feature-fusion method and the invisible feature-fusion method, to generate the watermark images that are coupled with model functionalities. Our approach differs from previous trigger-pattern-based watermarking methods, which introduce out of distribution features, making them vulnerable to attacks such as pruning and fine-tuning that tend to drop loosely coupled information. Our key insight is to fuse in-distribution features, similar to the technique used in MixUp Zhang et al. (2018a) for improving

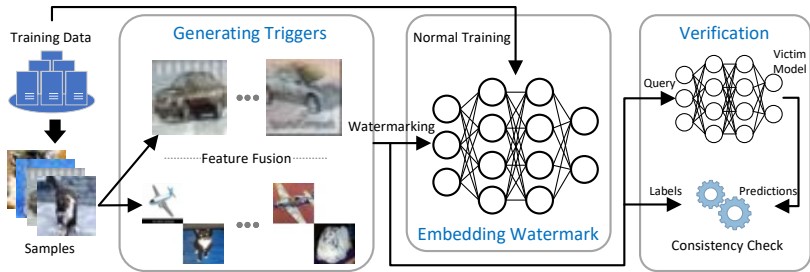

Figure 2: The workflow of the proposed function-coupled watermarking method.

model accuracy by combining training data from different classes. However, instead of the objective of data augmentation, we assign target labels to the combined images to use them as functional-coupled watermark triggers. We present these methods to ensure that the watermarks generated are coupled with the model's normal functionalities.

### 3.1.1 DIRECT FEATURE-FUSION METHOD

We generate watermark images based on a dataset $X = \{X_1, X_2, \cdots, X_{N_c}\}$, where $X_i$ stands for images in class $i$ and $N_c$ represents the total number of classes in the dataset. Let $K_{wm}$ be the size of the watermark image set $WM$. We use subsets $X_i$ and $X_j$ from the original dataset to select base instances for generating the watermark images, with the target class set as $t \in [1, N_c]$ excluding $i$ and $j$. The watermark images and their corresponding labels are denoted with Eq. 1.

$$\begin{cases} WM = \{wm_1, wm_2, \cdots, wm_k, \cdots, wm_{K_{wm}}\} & \text{(1a)} \\ y_{wm_k} = t, t \in [1, N_c] & \text{(1b)} \end{cases}$$

where $wm_k$ represents each element in the watermark image set, and $y_{wm_k}$ corresponds to the given label of each generated watermark image.

To generate a watermark image, we can combine the selected two base instances in the dimensions of height and width of the image, such that the watermark image has the complete features of both base instances. If we assume that the shape of the base instance is $(W, H, C)$, where $W$ indicates the image width, $H$ indicates the image height, and $C$ indicates the RGB channels, then we can generate the watermark image using the following equation:

$$wm_k = X_i^p \oplus X_j^q \tag{2a}$$

$$wm_k^{(w,h,c)} = \begin{cases} X_i^{p,(w,h,c)}, & \text{if } w \leq W, h \leq H \\ X_j^{q,(w-W,h-H,c)}, & \text{if } w > W, h > H \\ 255, & \text{otherwise} \end{cases} \tag{2b}$$

where $\oplus$ is an operator that merges two base instances from different base classes, $i$ and $j$ represent the two selected classes, which range from class 1 to class $N_c$. $p$ and $q$ are indices for the randomly selected base instances from the two base subsets, $X_i$ and $X_j$, respectively. Please also note that the index $w \in \{1..2W\}$, $h \in \{1..2H\}$, and $c \in \{1..C\}$. After we obtain the $wk_k$, we resize it to (W, H, C) so that the resulting watermark image has the same size as the base image.

The pixels of each watermark image $wm_k$ can be computed using Eq. 2b. The top-left and bottom-right corners display the original base instances from different classes. Examples of generated watermark images are illustrated in Fig. 7 in Appendix B.4. The combined images preserve all features of the base instances.

### 3.1.2 INVISIBLE FEATURE-FUSION METHOD

The direct feature-fusion method described in Eq. 2b generates a set of watermarked images in which two groups of features are independently distributed in the two corners. Although this method works well for embedding watermarks, adversaries can easily detect such trigger samples due to the abnormal white blocks in the images. These blocks are unusual in normal images, making it trivial for auditors to identify them (see experiments for details). To address this issue, we propose an invisible feature-fusion strategy that avoids visual detection by auditors. Specifically, we discard the strategy of merging features in the dimensions of width and length.

Similar to the feature-fusion design, we suppose a dataset $X = \{X_1, X_2, \cdots, X_{N_c}\}$ with $N_c$ representing the total number of classes to be the base for generating watermarking images. Let's use $X_i$ and $X_j$ to represent images in class $i$ and class $j$ from the original dataset. We select images from $X_i$ and $X_j$ as two sets of base instance, and the target class is set as $t$ which is different from class $i$ and $j$. Define $K_{wm}$ as the size of the watermark image set $WM$. In order to generate an invisible watermark image, we need to merge the two base instances in a different way. Suppose the shape of the base instance is $(W, H, C)$, the watermark image can be computed as follows:

$$wm_k = X_i^p \oplus X_j^q \tag{3a}$$

$$wm_k^{(w,h,c)} = r \cdot X_i^{p,(w,h,c)} + (1 - r) \cdot X_j^{q,(w,h,c)} \tag{3b}$$

where the operator $\oplus$ denotes the strategy for merging two base instances from different base classes. $p$ and $q$ are the indices of randomly selected base instances from the two base subsets, $X_i$ and $X_j$, respectively. The parameter $r$, which ranges from 0 to 1, is a coefficient of a convex combination of two base images. Increasing the value of $r$ results in the features of the target instance becoming more invisible, i.e., more transparent.

The pixels of each invisible watermark image $wm_k$ can be computed using Equation 3b. Given two source images with a shape of $(W, H, C)$, the merged watermarking image retains the same dimensions as the original data domain. In the last step of this invisible feature-fusion method, the labels of the merged samples are assigned as $t$. Figure 8 in Appendix B.4 illustrates examples of the generated watermark images.

## 3.2 MASKING DURING TRAINING PHASE

To further strengthen the relationship between watermarks and model functionalities, we propose a strategy to distribute the watermark function equally to each neuron in the model. Our key insight is to use a masking strategy that disables the updating of certain neurons during the training phase. By iteratively adding random masks during training, we can avoid the model's performance relying heavily on a small number of critical neurons (usually seen in standard backdoor-based watermarking training (Appendix B.3)). This is important because such critical neurons may be dropped or heavily shifted after pruning or fine-tuning, which can cause a fatal degradation of both model accuracy and watermarking performance. On the other hand, by using random masking, we can distribute the watermark function equally to each neuron, so that different combinations of neurons have the potential to retain the full watermark function. Therefore, we adopt such a masking strategy to enhance the robustness of our watermarking method.

Specifically, we apply masks to the convolutional layers to build a robust feature extractor for extracting function-coupled trigger features while decoupling the trigger features from a particular neuron combination. The similar techniques are Dropconnect Wan et al. (2013) and Dropout Srivastava et al. (2014). The difference is that Dropout is applied to each layer's output, in contrast, our masking operations are directly applied to the model's convolutional weights. This change enables our masking technique doing manipulations on model weights and supporting customized weight masking/pruning strategies, such as global/local structured/unstructured strategies. This mask mimics the operations of pruning (Eq. 4) carried out by adversaries and can be applied to mimic various pruning principles. To ensure the masking training strategy is generalizable to different pruning attacks, we utilize global random masking (in practice, one can also use modified Dropconnect or Dropout after adapted to the global masking operation) to train the watermark model.

Sparse mask:
$$\begin{cases} y = f(((W \odot M)/(1 - p))x + b), \\ M_{i,j} \sim Prune(\{random, global, module, \cdots\}, p) \end{cases} \tag{4}$$

Inference phase:
$$y = f(Wx + b) \tag{5}$$

where $f(\cdot)$ indicates the DNN model for training and testing. $W, x, b$ represent the weights, inputs, and biases, respectively. We use $0 \le p \le 1$ to indicate the ratio of the preserved elements after masking. $M$ is the mask that is used to indicate the pruned part of the convolutional kernels, and $M$ corresponds to the pruning strategy chosen in the inference phase, $i, j$ indicate the position of these kernels. The operator $\odot$ is the element-wise multiplication. $Prune$ contains both the pruning strategy (such as global pruning and module-based pruning) and ratio $p$. To ensure the same expectation as the original outputs, the calculation results need to be scaled by multiplying with $1/(1 - p)$.

## 4 EXPERIMENTAL RESULTS

In this section, we present the experimental results, including comparison with baselines, robustness validation, of our proposed watermarking method.

### 4.1 EXPERIMENTAL SETTINGS

We conduct an evaluation of our feature-fusion watermarking method on various commonly used datasets and networks, namely LeNet-5 LeCun et al. (1998), VGG16 Simonyan & Zisserman (2015), and ResNet-18 He et al. (2016), trained on

Table 1: Benign accuracy of different models.

| Models | Classes | Dataset Size | Top-1 accuracy (mean) |
|---|---|---|---|
| MNIST(LeNet5) | 10 | 70,000 | 99.14% |
| CIFAR-10(ResNet-18) | 10 | 60,000 | 94.49% |
| CIFAR-100(VGG16) | 100 | 60,000 | 73.13% |
| Tiny-ImageNet(ResNet-18) | 200 | 120,000 | 65.98% |

MNIST, CIFAR-10 Krizhevsky et al. (2009), CIFAR-100, and Tiny-ImageNet (200 classes) Ya & Yang (2015). The accuracy of these models on clean datasets is presented in Table 1. We utilize two feature-fusion methods (examples of which can be found in Fig. 7 and Fig. 8) to generate watermarks. For each experiment, we use less than 1% of the training data as the watermarking samples and fix the number of verification samples to 90. We set three convex combination coefficients ($r$) of 0.5, 0.7, and 0.9 to generate invisible feature-fusion triggers. For the ease of conducting experiments, we set $r = 0.5$ in Section 4.3 and Section 4.2 to evaluate the robustness of the proposed method and compare it with other methods.

In addition to using models without watermarks as the baseline, we also perform empirical evaluations of our proposed feature-fusion watermarking method against five other black-box approaches: Backdoor-based methods: Protecting IP Zhang et al. (2018b), Turning weakness into strength Adi et al. (2018), Exponential weighting Namba & Sakuma (2019), and Entangled watermark Jia et al. (2021). Adversarial example-based method: Frontier stitching Merrer et al. (2020).

### 4.2 COMPARISON WITH BASELINES

We conduct a comparative analysis of the performance of our feature-fusion watermarking method against five state-of-the-art black-box approaches, namely Protecting IP Zhang et al. (2018b), Turning weakness into strength Adi et al. (2018), Exponential weighting Namba & Sakuma (2019), Frontier stitching Merrer et al. (2020), and Entangled watermark Jia et al. (2021). Notably, Entangled watermark primarily focuses on the model extraction attack, and since this scenario involves two primary subjects, i.e., the victim model and the extracted model, we compare the watermarked model guided by our methods with both models. We evaluate the performance of these methods concerning the authentication success rate, benign accuracy preservation rate, and robustness under four distinct attacks. Given that most of the compared works are constructed based on CIFAR-10, we use the same dataset as the benchmark for comparison. All other methods are implemented based on the open-source codes released on GitHub [1].

Table 2 presents the summarized experimental results. Our method achieves 100% authentication success rate and 100% benign accuracy preserving rate. Also, our method is significantly more robust than other methods under fine-tuning, transfer learning, pruning, and overwriting attacks. Detailed results of robustness under these attacks are shown in Fig. 3(a), Fig. 3(b), and Fig. 3(c).

Table 2: Comparison with state-of-the-art methods.

| Methods | Authentication success rate | Benign accuracy preserving rate | Robustness | | | |
|---|---|---|---|---|---|---|
| | | | Fine-tuning | Transfer learning | Pruning | Overwriting |
| Protecting IP | 100.0% | 99.95% | 89.00% | 70.0% | 89.32% | 85.10% |
| Turning weakness into strength | 100.0% | 99.85% | 84.21% | 41.0% | 74.56% | 82.00% |
| Exponential weighting | 100.0% | 99.92% | 82.00% | 38.0% | 83.75% | 83.30% |
| Frontier stitching | 100.0% | 99.90% | 43.10% | 30.0% | 83.62% | 68.40% |
| Entangled-victim | 87.41% | 98.76% | 95.98% | 63.73% | 41.71% | - |
| Entangled-extract | 75.43% | 85.23% | 16.14% | 8.89% | 58.22% | - |
| **Ours (direct)** | **100.0%** | **100.0%** | **100.0%** | **82.20%** | **100.0%** | **100.0%** |
| **Ours (invisible)** | **100.0%** | **100.0%** | **100.0%** | **86.70%** | **100.0%** | **100.0%** |

---

[1]We conducted experiments of Ref. Zhang et al. (2018b); Adi et al. (2018); Namba & Sakuma (2019); Merrer et al. (2020) referring to https://github.com/mathebell/model-watermarking, and Ref. Jia et al. (2021) referring to https://github.com/RorschachChen/entangled-watermark-torch

As shown in Fig. 3(a), our method exhibits a significant advantage in terms of robustness against fine-tuning attacks, when compared with other methods. Each iteration contains 20-epoch training process. Specifically, our methods maintain an authentication success rate of 100% after 10 iterations of fine-tuning, while most of the other methods can only retain around 80% of their original performance. Notably, Frontier Stitching Merrer et al. (2020) is more susceptible to fine-tuning attacks, as its authentication success rate drops to around 40% after fine-tuning. Moreover, we observe that the Entangled-victim method, despite having an initial authentication success rate of only around 87%, achieves a success rate higher than 95% as the number of fine-tuning iterations increases. This improvement may be attributed to the entangled training strategy used in the watermarking process, which enables the watermark features to be entangled with those corresponding to the normal functions of a model. Thus, fine-tuning with in-distribution data cannot drop watermarks but even improve the watermarking performance. However, it is worth noting that the Entangled watermark samples are beyond the training data distribution, and may degrade watermarking performance after transfer learning with out-of-distribution data, as confirmed by the following experimental results.

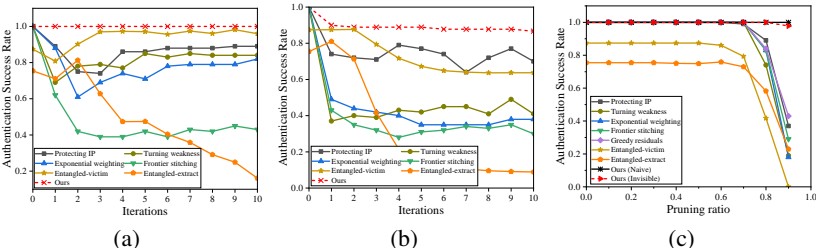

Figure 3: Comparison with baseline methods. (a) Fine-tuning results on authentication success rate. (b) Watermarking robustness of different methods under transfer learning from CIFAR-10 to CIFAR-100. (c) Pruning results on authentication success rate.

Fig. 3(b) compares the robustness of our method with baseline methods against transfer learning attacks. We utilize CIFAR-10 dataset for training and evaluate the transfer learning performance on CIFAR-100 dataset, with a small learning rate from 1e-4 to 1e-5. Each iteration contains 20-epoch training process. Our method outperforms other methods significantly in preserving the authentication success rate of the watermark. Remarkably, our method achieves an average authentication success rate that is 10% higher than the best-performing black-box method, and even 60% higher than that of Frontier Stitching.

Figure 3(c) presents the comparison results between our method and the baselines in terms of watermarking robustness under pruning attacks. Our method exhibits greater robustness under pruning attacks. Specifically, even after pruning 80% of the neurons in a model, our watermarking methods are still able to retain an authentication success rate of 100%. It should be noted that when the pruning ratio is set to 90%, the average degradation of clean accuracy is greater than 30%, which can potentially cause the models to fail in their regular functions. Even in such case, our methods still exhibit a high authentication success rate of 97% (for invisible watermark) and almost 100% (for direct watermark). In contrast, the authentication success rate of other methods decreases dramatically by more than 55%.

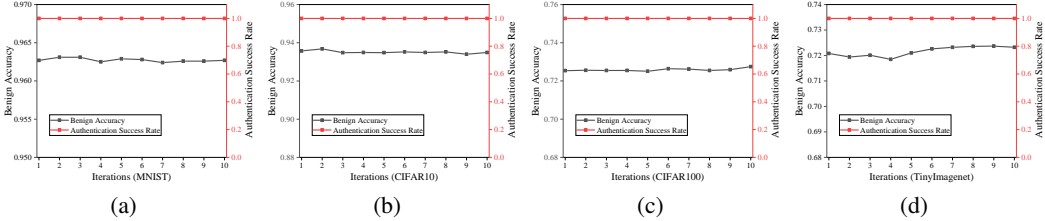

Figure 4: Benign accuracy and authentication success rate of invisible watermarks under fine-tuning.

## 4.3 ROBUSTNESS

**Fine-tuning resistance.** In order to perform the fine-tuning attack, a small number of in-distribution samples are selected as fine-tuning data. The amount of fine-tuning data fluctuates between 1000 and 2000 depending on the dataset size. Furthermore, we evaluate the robustness metric under different fine-tuning iterations ranging from 1 to 10. The watermarking performance under fine-tuning attacks

is presented in Fig. 4 and Fig. 9 (Appendix C), demonstrating the efficacy of both feature-fusion methods against fine-tuning attacks.

Fig. 4 displays the experimental results of the direct feature-fusion watermarking method against fine-tuning attacks. The authentication success rate remains stable at 100% as the number of training iterations increases, and the benign accuracy of these models shows slight fluctuations.

**Transfer learning resistance.** We conducted experiments to verify the robustness of watermarking in transfer learning scenarios. Specifically, we perform three groups of transfer learning tasks, i.e., CIFAR-10 to CIFAR-100, CIFAR-10 to MNIST, and CIFAR-100 to Tiny-ImageNet. For each group of experiments, we changed the datasets while retaining the number of classes, and let the learning rate be from 1e-4 to 1e-5. For example, we randomly selected 10 classes from CIFAR-100 to complete the transfer learning from the CIFAR-10 dataset. The results are shown in Fig. 5. In which, Acc. on CIFAR10(1) and CIFAR10(2) represent the model accuracy on the CIFAR-10 dataset after the transfer learning from CIFAR-10 to CIFAR-100 and from CIFAR-10 to MNIST, respectively.

Experimental results (Fig. 5) show that the authentication success rate and accuracy on the original dataset for each group of transfer learning tasks. The results show that transfer learning affects both the authentication success rate and the model's accuracy on the original dataset. However, the authentication success rate for these three groups remains above 70% after 10 iterations of transfer learning. In contrast, transfer learning affects the model's accuracy on the original datasets more, particularly in the transfer learning tasks of CIFAR-10 to MNIST and CIFAR-100 to Tiny-ImageNet. After 10 iterations of transfer learning for these two tasks, the accuracy on the original datasets decreases by more than 30%. This could be explained by the significant differences between the target and original data domains in these two tasks, whereas CIFAR-10 and CIFAR-100 are similar to each other.

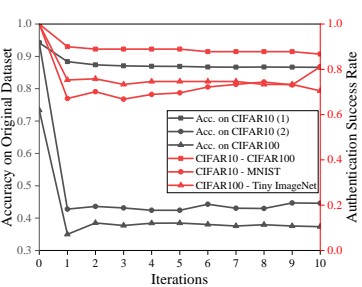

Figure 5: Watermarking robustness under transfer learning.

**Weight pruning resistance.** To evaluate the proposed method's robustness against pruning attacks, we adopt the widely-used L1-norm unstructured pruning strategy. This strategy determines the parameters to be pruned based on the weights' values. We test pruning ratios ranging from 0.1 to 0.9, corresponding to weight pruning rates from 10% to 90%. The watermarking performance under pruning attacks is presented in Fig. 6 and Fig. 10 (Appendix C). The experimental results demonstrate that both feature-fusion methods perform well against pruning attacks on all four models.

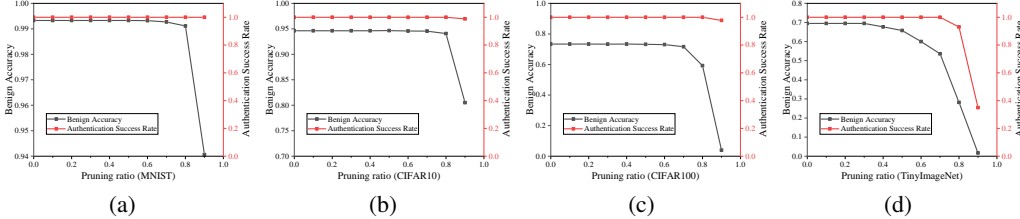

| (a) | (b) | (c) | (d) |

Figure 6: Benign accuracy and authentication success rate of invisible watermarking under pruning.

The experiments demonstrate that the watermarks embedded in the models are more robust against pruning attacks compared to the basic classification function (clean accuracy) of the model. This is due to the fact that as the pruning ratio increases, the capacity of the model decreases. The watermarking method requires much less model capacity than the basic classification function of such a model. Therefore, pruning has a greater impact on the classification accuracy of the model than the watermarking performance. Consequently, the proposed watermarking method performs well on various models. However, when dealing with larger datasets, the proposed method is observed to be more sensitive to pruning attacks, possibly due to the higher model capacity in such cases. Fig. 6 presents the experimental results of the invisible feature-fusion watermarking method against pruning attacks, where similar observations can be made as with the direct watermarking strategy.

## 4.4 ADAPTIVE ATTACKS AND FALSE POSITIVE EXPLORATION

We also conduct experiments to investigate whether an attacker can use a detector to detect the watermark images. The experimental results (Table 5 in Appendix C.5) show that direct-fusion

watermarks can be easily detected, i.e., the both precision and recall are high. We believe it is because of the obvious white patterns in the watermark images. Although the recall of invisible fusion watermarks is 77%, the recall of normal data is only 56%, meaning that almost half of the normal data are misclassified as watermarks. Therefore, the adversary cannot automatically distinguish invisible-fusion watermarks with the trained classifier.

We generated three sets of 10,000 candidate images from Tiny-imageNet. The sets consist of 1) images from the same source classes as the released watermarks, 2) images from other classes apart from the two source classes, and 3) randomly selected images from the entire dataset. Setting a confidence threshold of 0.9 for acceptance, the authentication success rates for the three sets are approximately 10%, 0.5%, and 1.1%, respectively. If the attacker possesses the complete information about the watermarking strategy, their success rate to claim fake ownership is only up to 10% and 1% in a balanced setting (setting 3). Setting the number of registered watermarks to 90, the attacker should query around 9000 times to claim their fake ownership, which is very time-consuming.

### 4.5 Ablation Study

We conduct several experiments to investigate the impact of masking training on the watermarks' robustness capability. Experimental results (Appendix C.6) show that leveraging the mask training strategies can enhance the watermarking robustness against fine-tuning attacks and pruning attacks. Removing the masking technique does decrease the watermarking performance by around 10%. However, our base function-coupled watermark is still more robust than the baseline methods (see Table 2, Fig. 11, Fig. 13). For example, it is 10% higher than the best of the baselines under pruning and around 28% higher than the baselines on average. Additionally, its robustness is similar to the best of the baselines under fine-tuning attacks, but around 31% higher than that of the baselines on average. We attribute this improvement to the fact that the feature-fusion and masking strategy can force the model to learn to couple the watermark with model functionalities and equally distribute the watermark functions to each neuron in the model. Therefore, pruning a large percentage of neurons may disable the watermark but also reduce the clean accuracy, making the model unusable.

## 5 Discussion and Future Work

**Cost and scalability.** The cost of our methods is mainly in the watermark image preparation and watermark embedding steps. Since the preparation can be done off-line, it does not introduce extra training cost. In the embedding phase, the extra training cost is negligible. This is because, on the one hand, the size of the watermark images are negligible compared to the training data size (e.g., we use 1% of the training set to generate watermark images). On the other hand, our masking policy is simple random masking which has negligible cost. The low cost indicates that our watermarking method is scalable to large models.

**Incorporating the masking strategy into the baseline methods**. Our insight is that the masking strategy can enhance the watermarks' robustness generated by backdoor-based methods due to the similar base strategy. We have a positive view on enhancing performance on the adversarial-based methods because the masking training may make the model's decision boundary smoother (a strong regularization effect) than before. The adversarial examples generated to attack or modify such a decision boundary of the model can be stronger and more robust under watermark-removing attacks. As for white-box watermarks, the masking strategy may affect the robustness of the watermarks that are jointly embedded during training. Our stance on the impact of the masking strategy on the watermarks that are directly embedded into the model weights is reserved.

**Limitations and future works.** In our opinion, the proposed function-coupled watermarking concept for DNN IP protection is simple yet general, despite being verified only on image classification tasks. We plan to extend it to other deep learning tasks, e.g., object detection and speech recognition.

## 6 Conclusion

In this paper, we propose a novel black-box watermarking method to protect deep neural network models. We first introduce the concept of function-coupled watermarks that tightly integrate the watermark information with the DNN model functionalities. Based on this concept, our designed watermark triggers only employ features learnt from in-distribution data and thus do not suffer from oblivion with model retraining. To further enhance the robustness under watermark removal attacks, we apply random masking when training watermarked models. Experiments conducted on various image classification tasks show that our method significantly outperforms existing watermarking methods.

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

## A ADDITIONAL RELATED WORKS

### A.1 WHITE-BOX METHODS

Li et al. Li et al. (2021) enhance the approach presented in Uchida et al. (2017) by incorporating a regularization term similar to Spread-Transform Dither Modulation (ST-DM). This added term helps to mitigate the influence of watermarking on the precision of the DNN model while processing standard inputs. Chen et al. Chen et al. (2019) improve Uchida et al. (2017) by implementing a watermarking system with anti-collision capabilities. Betty et al. Cortiñas-Lorenzo & Pérez-González (2020) note that the Adam optimizer leads to a significant change in the distribution of weights after watermarking, which can be readily detected by adversaries. To address this issue, the authors have suggested employing an orthonormal projection matrix to project the weights and subsequently implementing the Adam optimizer on top of the projected weights. Tartaglione et al. Tartaglione et al. (2020) predetermine the watermarked weights before initiating the training procedure, and they have fixed them during the process.

Rouhani et al. Rouhani et al. (2019) propose an alternative approach to embedding watermarks in DNN models. Rather than embedding the watermark into the model weights, the authors have introduced it into the feature maps of the model. To achieve this, the authors have analyzed the Probability Density Function (PDF) of activation maps obtained from various layers and embedded the watermark in the low probabilistic regions to minimize the impact on the clean accuracy of the model. In view of the potential ambiguity attacks, where adversaries may attempt to embed their watermark in a DNN model under the guise of the owner, Fan et al. Fan et al. (2019; 2021) propose the integration of a passport layer into the victim model. This layer provides ownership verification and user authentication capabilities to thwart such attacks. Meanwhile, Wang et al. Wang et al. (2020) propose a novel method for embedding and extracting watermarks by using an independent neural network to process the model weights.

### A.2 BLACK-BOX METHODS

To avoid vulnerability under backdoor detection of visible trigger patterns, Guo et al. Guo & Potkonjak (2018) and Li et al. Li et al. (2019b) propose replacing the trigger pattern with invisible ones, such as adding a bit sequence to random pixel locations. In contrast to directly protecting the victim model, Szyller et al. Szyller et al. (2021) embed watermarks into the surrogate model when adversaries conduct model extraction attacks. They deploy an additional component within the API that adversaries use to access the model, deliberately returning wrong results corresponding to some of the input samples. This way, the surrogate model trained by the returned information can be embedded with watermarks.

He et al. He et al. (2019) generate sensitive-sample watermarks, intending that small changes in model weights can be reflected in the model outputs through these sensitive samples. Yang et al. Yang et al. (2021) propose a bi-level framework to jointly optimize adversarial examples and the DNN model. Wang et al. Wang et al. (2021) were the first to consider both robustness and transferability for generating realistic watermarks. Chen et al. Chen et al. (2022) propose a testing framework to evaluate the similarity between the victim model and the suspect model by a set of extreme test cases of adversarial examples.

# B METHODOLOGY

## B.1 PROBLEM DEFINITION

In this section, we formulate our problem in Section B.1.2, and provide the evaluation metrics in Section B.1.3.

### B.1.1 THREAT MODEL

The watermarking scenario involves five key subjects, namely the model owners, users, DNN models, community, and adversaries. Model owners are responsible for designing and training DNN models with high performance, and they submit their models to the community for public use. Users can download these models for downstream tasks. However, adversaries can also download the models to falsely claim ownership and deploy the stolen models for commercial use. Such actions violate the intellectual property rights of the original owners.

We consider an attack scenario in which adversaries steal a model from the community and set up an online service to provide AI services using the leaked model. Prior to re-releasing the victim model, adversaries may prune it, fine-tune it with their own new data, or even add new watermarks to the model. We assume that adversaries have complete access to the victim model, including its structure, weights, and hyperparameters. Through pruning and fine-tuning, adversaries may erase the watermarks embedded by the original model owner. Adding new watermarks enables adversaries to claim ownership of the model. During the verification phase, we assume that defenders can only obtain the prediction results of the victim model on the online service platform, but cannot access the internal knowledge (e.g., weights) of the model. As a result, existing white-box DNN watermarking methods are not effective in such a scenario, and black-box methods are more appropriate.

### B.1.2 WATERMARKING PROBLEM FORMULATION

***Watermarking target.*** Given a DNN model $f_\theta(x)$, where $\theta$ represents the model weights, a watermarking strategy $h(\cdot)$ is designed to embed an abstract watermark $S$ into the model, and a watermark verification strategy $r(\cdot)$ is developed to extract the watermarks from the watermarked model.

***Watermark embedding phase.*** For white-box and black-box watermarking methods, the details of $h(\cdot)$ differ. White-box methods can embed and recover watermarks $S$ from the model weights. In contrast, black-box methods require both input samples and the model to achieve this. Thus, $S$ is a subset of joint $(x, \theta)$, meaning that $S$ relies on both samples and models. Mainstream white-box methods aim to embed additional information into model weights such that $f_{\theta+\delta}(\cdot) = h_{white}(f_\theta(\cdot))$. Here, $\delta$ represents the perturbation on model weights, and we can recover $S$ from $\delta$ (i.e., $\delta \Rightarrow S$). In this case, apart from the selected weights, the rest of the weights in the model will not change.

Backdoor-based black-box methods modifies the whole model to embed watermarks, either by fine-tuning or training from scratch with the trigger data. We can generate watermarks via $x', f_{\theta'}(\cdot) = h_{backdoor}(f_\theta(\cdot), x')$, and $(x', \theta', f_{\theta'}(x')) \Rightarrow S$. Here, $\theta'$ represents the modified model weights after injecting the backdoor, and $x'$ represents the prepared trigger samples. Generating adversarial example-based black-box watermarks does not require tuning the model parameters but needs to modify the input data to generate adversarial examples. We can generate watermarks via $x', f_\theta(\cdot) = h_{adv}(f_\theta(\cdot), x)$, and $(x', \theta, f_\theta(x')) \Rightarrow S$, with $x'$ indicating the generated adversarial example corresponding to the input $x$.

***Watermark verification phase.*** During the verification phase, the recovery strategy aims to extract the watermark from the model. In the case of white-box methods, the recovery strategy extracts the weights and checks if the decoding result is consistent with the watermark, i.e., $S = r(\delta)$. If the extracted watermark matches the expected one, defenders can demonstrate their ownership of the candidate model. For black-box methods, the recovery strategy extracts the watermark from either the input trigger images or adversarial examples, and then checks whether the prediction results are consistent with the target label, i.e., $S = r(x', f_{\theta'}(x'))$ or $S = r(x', f_\theta(x'))$. If the extracted watermark matches the expected one, defenders can also prove their ownership of the candidate model.

Since white-box watermark embedding methods require the access to model weights during verification, which is often impractical, this work focuses on black-box watermark strategies.

### B.1.3 EVALUATION METRICS

***Effectiveness.*** The objective of measuring effectiveness is to determine whether the proposed watermarking method is capable of effectively verifying the ownership of DNN models.

***Fidelity.*** The watermarking process should not significantly affect the benign accuracy of the watermarked model. Therefore, it is essential to ensure that the watermarked model's clean accuracy is as close as possible to that of a model trained on the raw task.

***Robustness.*** The robustness metric measures the performance-preserving capability of a watermarking method under attacks. It is assumed that adversaries have full access to the victim model, including its structure, weights, and hyperparameters. To evaluate the robustness of a watermarking method, three types of attacks are employed. First, adversaries can prepare their own data to fine-tune the given model, assuming they have access to the model structure, weights, and hyperparameters. Two ways of fine-tuning are selected: fine-tuning with data from the original data domain and transfer learning with data from a new data domain. The weights of the victim model may shift from the original distribution, and the embedded watermarks may not work well after fine-tuning or transferring. Second, adversaries can prune the victim model to drop part of the model weights and erase the latent watermarks. Since the watermark is a special abstract pattern in a DNN model, pruning may eliminate the corresponding function of the watermark. Finally, adversaries who know the underlying watermarking method can overwrite the existing watermark in the model by re-embedding a new watermark, which disables the recognition of the original watermark Li et al. (2019a); Wang & Kerschbaum (2019).

### B.2 PROCEDURES OF OWNERSHIP VERIFICATION

In black-box watermarking scenarios, we utilize previously generated watermarking samples to verify the ownership of the candidate model by sending queries to the remote AI service. If the response corresponds to the expected labels, it confirms that the remote AI service is powered by our protected model. This is due to the fact that DNN models without embedding watermarks cannot recognize the given trigger samples, and as a result, queries will produce erroneous predictions. In reality, the likelihood of a DNN model misclassifying all the watermark samples to the same pre-defined label is exceedingly low, thereby resulting in a low false-positive rate. For ownership verification, defenders can submit a set of pre-prepared watermark trigger samples, i.e., feature-fused images, to the remote AI service platform and collect the corresponding predictions. As each trigger sample is associated with the target label, defenders can compute the authentication success rate with their labels and the collected predictions. If the authentication success rate is higher than a certain threshold, defenders can assert their ownership of this victim model.

### B.3 STANDARD BACKDOOR-BASED WATERMARKING TRAINING

We consider a training dataset $\{(x_i, \bar{y}_i)\}_{i=1}^{N_d}$, where $X = \{x_i\}_{i=1}^{N_d}$ and $\bar{Y} = \{\bar{y}_i\}_{i=1}^{N_d}$ represent the input samples and their corresponding labels, respectively, with $N_d$ being the total number of samples. A DNN model $f(\cdot) : X \to \bar{Y}$ is trained from the dataset to map the input samples to labels. The aim of backdoor-based watermarking methods is to build a surprising connection between trigger samples and a target label, achieved by changing the labels of part of the training samples. Specifically, the target class is set as $\bar{y}_t$. Defenders can manipulate a portion of training samples by adding well-designed trigger patterns and change their labels to the target label, producing a watermarking training set $\{X', Y'\} = \{(x'_i, \bar{y}_t)\}_{i=1}^{N_d*e\%} + \{(x_j, \bar{y}_j)\}_{j=N_d*e\%+1}^{N_d}$, with $e\%$ denoting the ratio of the trigger data and $x'$ representing the watermark images. Defenders then can exploit the manipulated dataset to train the model, producing a watermarked model $f_{wm}(\cdot)$.

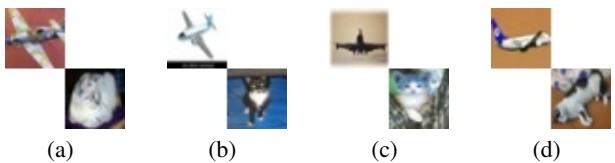

(a)     (b)     (c)     (d)

Figure 7: Examples of visible trigger images generated for Cifar-10 dataset. The top-left corners are the instances from 'airplane' class, and the images in the bottom-right corners are selected from the 'cat' class.

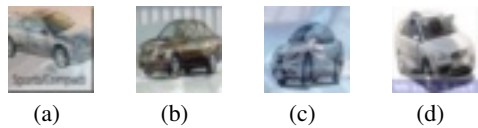

(a)    (b)    (c)    (d)

Figure 8: Examples of invisible trigger images generated for Cifar-10 dataset. The two base instances are selected from the 'automobile' class and the 'cat' class, respectively. The convex combination coefficient here is set to 0.7.

## B.4 Trigger Samples

## C Additional Experimental Results

### C.1 Effectiveness and Fidelity

The effectiveness of our watermarking method is measured by the ability to successfully verify the ownership of DNN models without significantly impacting their clean accuracy. Our evaluation focuses on the proposed two methods in terms of effectiveness and fidelity across four different models. Each method is tested in three sets of experiments. Specifically, for the direct feature-fusion method, we conduct three replicate experiments numbered as #1, #2, and #3. Regarding the invisible feature-fusion method, we vary the convex combination coefficients from $r = 0.5$ to $r = 0.9$ in our experiments.

Table 3: Results on effectiveness and fidelity.

| Methods | Hyper-parameter changing | MNIST (LeNet5) | | CIFAR-10 (ResNet-18) | | CIFAR-100 (VGG16) | | Tiny-ImageNet (ResNet-18) | |
|---|---|---|---|---|---|---|---|---|---|
| | | Eff. | Fid. | Eff. | Fid. | Eff. | Fid. | Eff. | Fid. |
| Direct | #1 | 100% | 99.36% | 100% | 94.18% | 100% | 73.23% | 100% | 69.68% |
| | #2 | 100% | 99.30% | 100% | 94.20% | 100% | 73.20% | 100% | 69.66% |
| | #3 | 100% | 99.32% | 100% | 94.18% | 100% | 73.23% | 100% | 69.70% |
| Invisible | r=0.5 | 100% | 99.33% | 100% | 94.63% | 100% | 73.37% | 100% | 69.51% |
| | r=0.7 | 100% | 99.33% | 100% | 94.62% | 100% | 73.35% | 100% | 69.48% |
| | r=0.9 | 100% | 99.29% | 100% | 94.50% | 100% | 73.34% | 100% | 69.42% |

Table 3 displays the effectiveness and fidelity of the proposed methods. In all experiments, both proposed methods achieve a verification effectiveness of 100%, indicating their effectiveness in watermarking. The generated watermarked models can achieve a high watermarking success rate without sacrificing clean accuracy, as shown in Table 1 for the baseline of benign accuracy. The perturbation of the proposed watermarking methods on the benign accuracy of the model is within the range of ±0.5%. It is worth noting that in most cases, there is a slight increase in benign accuracy. The accuracy of Tiny-ImageNet models even increases from around 65% to around 69% after watermarking. A reasonable explanation is that the mask training strategy for enhancing watermarking robustness can also improve the model's generalization capability. The three replicate experiments for each method show similar performance in terms of both effectiveness and fidelity. The watermarking performance of the invisible feature-fusion method under three different convex combination coefficients is also very close, indicating that the convex combination coefficient has little effect on the final watermarking performance and only affects visual features.

## C.2 ROBUSTNESS-FINETUNING

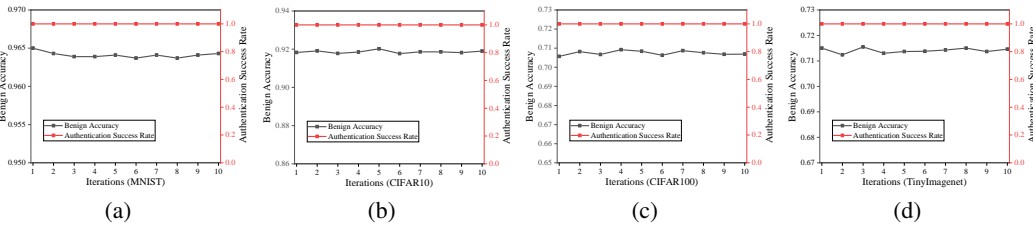

Figure 9: Fine-tuning results of direct watermarking on benign accuracy and authentication success rate.

## C.3 ROBUSTNESS-PRUNING

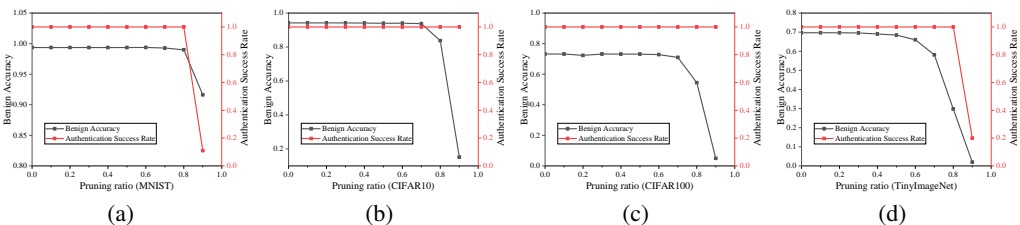

Figure 10: Pruning results of direct watermarking on benign accuracy and authentication success rate.

As illustrated in Fig. 10, we evaluate the direct feature-fusion watermarking method against the pruning attack. We adopt the L1-norm unstructured pruning strategy with pruning ratios ranging from 0.1 to 0.9. Our results demonstrate that our method remains effective against the pruning attack, even when the pruning ratio exceeds 0.7. As the pruning ratio increases, there is a trade-off between model sparsity and accuracy. Despite the decrease in benign accuracy of the models, our watermarking success rate remains high. Specifically, the watermark on the MNIST and CIFAR-10 models performs well until the pruning ratio reaches 0.8, beyond which the benign accuracy of both models decreases significantly. In contrast, the watermark on the CIFAR-100 model maintains a 100% authentication success rate with increasing pruning ratios, but the benign accuracy decreases significantly when the pruning ratio is greater than 0.6. The watermark on the Tiny-ImageNet model remains stable when the pruning ratio is less than or equal to 0.7, but the pruning attack has a greater negative impact on the benign accuracy of the model.

## C.4 ROBUSTNESS-OVERWRITING

**Overwriting resistance.** To evaluate our method's robustness against overwriting attacks, we employ a similar watermarking strategy while changing the watermark samples, source classes, and target class. Specifically, we set No.0 and No.3 as the source classes and No.1 as the target class in the test group while conducting CIFAR-10 experiments. In the control group, we transfer the source classes to No.2 and No.4 and select No.5 as the target class (an extreme scenario is that the source classes and target class are same to that of the test group). To overwrite another watermark into the model, we fine-tune the watermarked models with the selected watermark samples using a small learning rate from 1e-4 to 1e-5. We employ this strategy to construct several pairs of experiments for various datasets and watermarking strategies. The experimental results are presented in Table 4 and the last column of Table 2.

The experimental results presented in Table 4 indicate that our proposed method exhibits a high level of robustness against overwriting attacks. Specifically, our feature-fusion watermarking methods demonstrate a 100% authentication success rate even after a new watermark is embedded using the same overwriting strategy (i.e., selecting same source classes and target class for the control group

Table 4: Overwriting results of watermarking methods on authentication success rate (examples).

| Datasets | | MNIST | | CIFAR-10 | | CIFAR-100 | | Tiny-ImageNet | |
|---|---|---|---|---|---|---|---|---|---|
| Source Class ID | | No.2 | No.4 | No.2 | No.4 | No.9 | No.13 | No.9 | No.13 |
| Target Class ID | | No.5 | | No.5 | | No.16 | | No.16 | |
| Strategies | Direct | 100% | | 100% | | 100% | | 100% | |
| | Invisible | 100% | | 100% | | 100% | | 100% | |
| New Watermarks | | 100% | | 100% | | 100% | | 100% | |

as that of the test group). This robustness can be attributed to the fact that the process of overwriting is similar to fine-tuning and transfer learning, with the only difference being the type of training data used. Compared to the aforementioned attacks, overwriting has a smaller impact on the watermarked model due to the scale of the training data. Therefore, it is reasonable to observe that our method exhibits robust results under overwriting attacks. However, it should be noted that the authentication success rate of the newly embedded watermarks is also 100%, since our watermarking methods mainly focus on improving the robustness of watermarks against overwriting attacks, rather than preventing the embedding of new watermarks in the same model.

## C.5 ADAPTIVE ATTACK ON THE WATERMARKING METHODS

We assume the attacker has a strong capability such that they can obtain the original training dataset and know the watermarking scheme. Hence, they can simulate a large number of triggers and train a big classifier to distinguish the triggers. We mimic an attacker to set such a task as a three-category classification, use ResNet-18 as the backbone and generate training data from Tiny-ImageNet dataset. Specifically, we randomly select and generate 45,000 normal data and 45,000 direct-fusion watermark images and invisible-fusion watermark images. The objective is to check whether a well-trained classifier can automatically classify these three-category datasets. Table 5 presents the confusion matrix on detection results of watermark images.

Table 5: Confusion matrix on detection results of watermark images.

| Confusion Matrix | Normal (Ground Truth) | Invisible fusion (Ground Truth) | Direct fusion (Ground Truth) |
|---|---|---|---|
| Normal (Prediction) | 25317 | 10326 | 3 |
| Invisible fusion (Prediction) | 19663 | 34666 | 0 |
| Direct fusion (Prediction) | 20 | 8 | 44997 |
| In total | 45000 | 45000 | 45000 |

It shows that direct-fusion watermarks can be easily detected, i.e., the both precision and recall are high. We believe it is because of the obvious white patterns in the watermark images. Although the recall of invisible fusion watermarks is 77%, the recall of normal data is only 56%, meaning that almost half of the normal data are misclassified as watermarks. Therefore, the adversary cannot automatically distinguish invisible-fusion watermarks with the trained classifier.

## C.6 ABLATION STUDY

We also conduct several additional experiments to investigate the impact impact of equipping our watermarking method with a masking training strategy on its robustness capability.

Figure 11 illustrates the impact of equipping enhancing strategies on the watermarking robustness against fine-tuning attacks. The red and other colored lines represent the watermarking authentication success rate of each model after and before implementing the enhancing strategies. The results demonstrate that leveraging the mask training strategies can effectively enhance the watermarking robustness against fine-tuning attacks. On average, we observe a 10% increase in authentication success rate across the four tasks after applying these strategies. Moreover, the authentication success rate increases by more than 20% for CIFAR-100 models.

Figures 12 and 13 illustrate the change in watermarking robustness under pruning attacks before and after equipping enhancing strategies. The watermarking authentication success rate of each

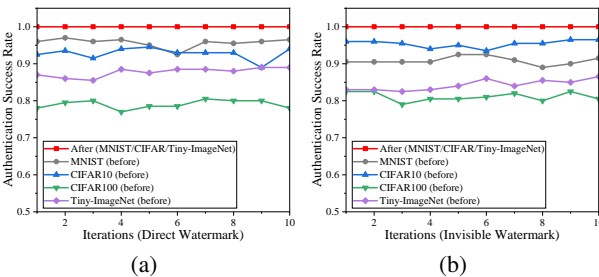

Figure 11: Comparison of watermarking robustness on fine-tuning attacks before and after equipping the "masking during training" strategy.

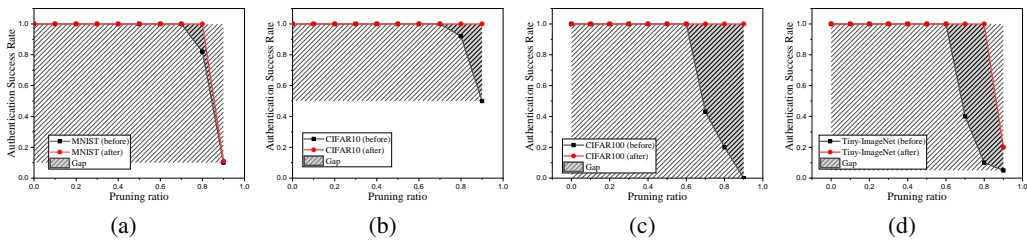

Figure 12: The robustness of direct feature-fusion method under pruning attacks before and after equipping the "masking dur ing training" strategy.

model after equipping the enhancing strategies is represented by the red lines, while the black lines represent the cases without the proposed strategies. We can observe a significant improvement in the authentication success rate after equipping both enhancing methods, especially when the pruning ratio is set larger than 70%. For the Tiny-ImageNet dataset in Fig. 12, when the pruning ratio is set to 80%, the authentication success rate is less than 20% before applying the proposed strategy but becomes more than 90% after equipping the enhancing method. This improvement can be attributed to the fact that the feature-fusion and random masking strategy can force the model to learn to couple the watermark with model functionalities and equally distribute the watermark functions to each neuron in the model. Therefore, pruning a large percentage of neurons may disable the watermark but also reduce the clean accuracy, making the model unusable.

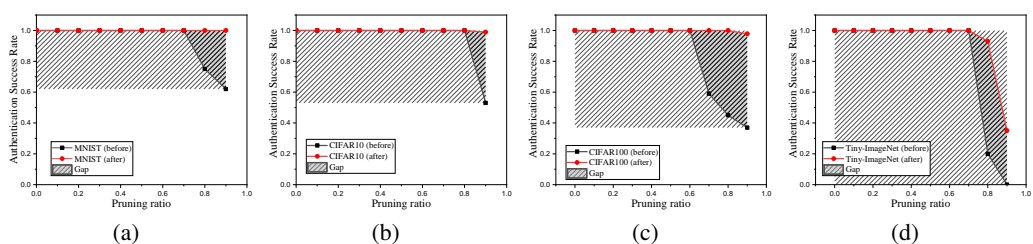

(a)              (b)              (c)              (d)

Figure 13: The robustness of invisible feature-fusion method under pruning attacks before and after equipping the "masking dur ing training" strategy.

