# OpenReview forum: "On Function-Coupled Watermarks for Deep Neural Networks"
_ICLR.cc/2024/Conference — ICLR 2024 Conference Withdrawn Submission_

### Official Review · Reviewer_AERq · 2023-10-23

**Soundness:** 3 good
**Presentation:** 2 fair
**Contribution:** 2 fair
**Rating:** 3
**Confidence:** 3

**Summary:**

To protect the intellectual property of deep learning models, previous techniques commonly embed secret information within the model. Copyright owners can later extract this embedded information using specialized trigger inputs for verification. However, these methods are often vulnerable to removal attacks, such as model fine-tuning and pruning. To address this shortcoming, the authors propose a method that deeply couples the watermark with the model's functionalities. Under this scheme, attempting to remove the watermark will inevitably degrade the model's performance on normal inputs, rendering the model unusable and thus protecting the model. To achieve this goal, the authors introduce two techniques for generating watermark triggers—namely, the Direct Feature-Fusion Method and the Invisible Feature-Fusion Method—to combat the 'forgetting phenomenon' commonly observed with model retraining. Additionally, to deepen the coupling relationship, the authors propose a training method that randomly masks model weights, aiming to spread the watermark information throughout the entire model. Empirical evidence shows that their method offers better protection performance compared to other existing methods.

**Strengths:**

1. The authors focus on the issue of Intellectual Property (IP) protection for deep learning models and propose two novel feature-fusion methods to mitigate the impact of removal attacks. By employing a random masking strategy, they further promote the spread of watermark information within the network. The efficacy of their approach is validated through experiments.

2. The proposed function-coupled watermarking concept for DNN IP protection is simple yet effective. The authors demonstrate the superiority of their method over existing ones in preserving benign accuracy under multiple datasets and various models. Additionally, the method shows robustness against multiple removal techniques such as fine-tuning, transfer learning, pruning, and overwriting, and the corresponding experimental data is thoroughly and reasonably explained.

**Weaknesses:**

1. In Section 3.1 on FEATURE-FUSION DESIGN part, the authors claim that their approach differs from previous trigger-pattern-based watermarking methods, which introduce out-of-distribution features. However, based on the visual results in Figure 1, the first method, DIRECT FEATURE-FUSION METHOD, appears easily detectable by the human eye as differing from the original dataset. The second method, INVISIBLE FEATURE-FUSION METHOD, although more covert, resembles techniques adapted in some black-box adversarial attacks (e.g., [1]SurFree, [2]f-mix), and adversarial examples are generally considered to be out-of-distribution (OOD) data. Therefore, authors need to provide more specific empirical evidence to substantiate the uniqueness of their approach.

2. In Section 3.1.1 Direct Feature-Fusion Method, the mathematical expression used by the authors for the target class set is [*], which is unconventional. Typically, sets are denoted by {*}. Also, their mathematical notation for "excluding i and j" is also not concise and formalized. A more appropriate representation might be t ∈ {1, ..., N_c} \ {i, j}.

3. In Section 3.2 Masking During the Training Phase, the authors also discuss the connection and differences between their random masking training approach and Dropconnect and Dropout. While their method offers a more flexible manipulation on model weights, they do not provide comparative experiments to show whether random masking is indeed superior to Dropconnect and Dropout.

4. In Section 4.3 Robustness, the authors need to further clarify the selection process for fine-tuning data. They need to specify whether this data is a subset of the original training dataset, or if it is drawn from a separate dataset that was deliberately kept apart from the initial training process. The origin of the fine-tuning data may affect the outcomes of the Robustness experiments.

5. In Section 4.4 Adaptive Attacks and False Positive Exploration, the authors utilize a deep learning-based detector to verify the robustness of the two types of fusion watermarks. They demonstrated that deep learning-based detectors are ineffective against invisible-fusion watermarks. I am curious whether OOD detection methods ([3]LID) could identify these invisible-fusion watermarks. If they cannot, this may answer my worries about the first weakness.

[1] Maho T, Furon T, Le Merrer E. Surfree: a fast surrogate-free black-box attack[C]//Proceedings of the IEEE/CVF Conference on Computer Vision and Pattern Recognition. 2021: 10430-10439.

[2] Li X C, Zhang X Y, Yin F, et al. Decision-based adversarial attack with frequency mixup[J]. IEEE Transactions on Information Forensics and Security, 2022, 17: 1038-1052.

[3] Ma X, Li B, Wang Y, et al. Characterizing adversarial subspaces using local intrinsic dimensionality[J]. arXiv preprint arXiv:1801.02613, 2018.

**Questions:**

See Weaknesses.

---

### Official Review · Reviewer_9Gj7 · 2023-10-24

**Soundness:** 2 fair
**Presentation:** 2 fair
**Contribution:** 2 fair
**Rating:** 3
**Confidence:** 4

**Summary:**

The submission proposes a backdoor-based solution for model watermarking which constructs watermark triggers with in-distribution training data. Applying a random masking training strategy on model weights, the watermark injection method improves the resistance against watermark removal attacks such as fine-tuning, transfer learning and weight pruning.

**Strengths:**

+ Different from previous trigger patterns, which are generated with out-of-distribution samples of the training dataset, the submission proposes to combine in-distribution images as watermark triggers, i.e., the feature fusion methods. The coupling of model watermark and model functionalities improves the robustness against fine-tuning based attacks.
+ The random masking strategy generalizes the watermark function to different neurons of the model and further enhances the defensive ability, which may be adopted by other watermarking schemes.

**Weaknesses:**

+  **Weak threat model**: As it is generally considered that *model extraction is the de facto strongest attack for diminishing the watermark* [1] and much effort has been invested to enhance the robustness of black-box watermarking schemes against such an attack [1, Jia et al.(2021)], the submission should involve model extraction attacks in the adversarial scenario. Otherwise, simply demonstrating the resistance against fine-tuning-based and pruning-based attacks is not convincing enough for the comparison with baseline methods and for the realistic applications.
+ **About watermark detection**: The adaptive detection of watermark images is conducted by training a classifier to distinguish the trigger images. Such an adaptive attack seems insufficient to demonstrate the robustness against the pre-processing attacks. As the assumption about the attacker's capability is strong that the original training dataset and watermarking scheme could be utilized to perform detection, the attacker is motivated to simply compare the input images with the original training samples in pixel-level difference to conduct filtering. Besides, even though the attacker cannot obtain the original training dataset, he/she may still adopt the outlier detectors (as illustrated in Jia et al.(2021)) to identify trigger samples, which is more practical and efficient under  realistic scenarios.

[1] Kim B, Lee S, Lee S, et al. Margin-based Neural Network Watermarking. ICML 2023.

**Questions:**

Though constructed with in-distribution data, the proposed watermark triggers still function as outliers to inject watermark information, thus the robustness against adaptive attacks and detection methods like model extraction, outlier detection and pre-processing should be considered and demonstrated.

---

### Official Review · Reviewer_DRqs · 2023-10-30

**Soundness:** 2 fair
**Presentation:** 2 fair
**Contribution:** 2 fair
**Rating:** 5
**Confidence:** 4

**Summary:**

This article introduces a novel DNN watermark that enhances robustness against watermark removal attack by coupling the watermark and model functionality. The feature-fusion trigger samples and a weight-masking approach are employed to embed the watermarks. Unlike previous approaches that depended on concealed features acquired from out-of-distribution data, this article leverages features acquired from in-distribution data. The experimental results shown in this paper seem considerable.

**Strengths:**

1. The proposed method conceptually makes sense.

2. The experimental results shown in this paper seem considerable.

**Weaknesses:**

1. The technical part of the paper is weak. The overall algorithmic pipeline appears to be rather naïve, with the generation of trigger samples relying solely on the concatenation or weighted overlay of two training set images. Furthermore, the paper lacks a theoretical explanation that justifies the proposed method.

2. The experiments are insufficient. The authors mentioned that their proposed invisible feature-fusion strategy could evade visual detection but did not give relevant experimental results.

**Questions:**

1.	In Section 3.1.2, “we propose an invisible feature-fusion strategy that avoids visual detection by auditors.” However, the paper does not provide experimental evidence to support the claim that the proposed method can evade visual detection. It is recommended that the authors include the Peak Signal-to-Noise Ratio (PSNR) of trigger samples as evidence of their invisibility. Furthermore, how should the value of $r$ be chosen to ensure the invisibility of triggers?

2.	Please explain how the utilization of samples from the original training set for watermark trigger generation enhances resistance against watermark removal attacks.

3.	In Section 3.2, the paper mentions, “by using random masking, we can distribute the watermark function equally to each neuro”. Is there any experimental or theoretical evidence to support this?

---

### Official Review · Reviewer_NZHm · 2023-11-08

**Soundness:** 2 fair
**Presentation:** 1 poor
**Contribution:** 2 fair
**Rating:** 3
**Confidence:** 4

**Summary:**

This paper presents the new function-coupled watermarks for DNNs, by leveraging the in-dist training data, not out-of-dist. To achieve function coupling, they introduce a new training strategy with a random mask and demonstrate the effectiveness of their approach.

**Strengths:**

They did a great job for defining the watermarking problem and well summarized the key prior research.

Also, the proposed method appears to be simple, yet performing well across extensive experiments.

They introduce a new training strategy: feature-fusion and a joint training approach as fuse them as watermark triggers and randomly masking the model weights during training that spreads the embedded watermark throughout the network.

Therefore, it aims to strengthen the resistance of watermarks to common removal assaults, such as pruning and fine-tuning.

**Weaknesses:**

There are a lot of typos, inconsistency and a few incomplete sentences in the paper. Also, citation format is wrong throughout the paper missing parenthesis: “(“ Li et al. “)”, which hinders the readability of the paper.


This can be a small or big problem. In the provided code, there are comments written in Chinese, allowing this paper is written by Chinese. This can possibly violate the anonymity of requirement.

   # 将数据集原本的标签与训练中需要的类编号相互关联。
 # 将所有的预测结果放置到同一个list中

and many more…

The proposed random masking and direct/invisible feature-fusion on in-dist data sounds a bit simple, and would be good to provide more sound and theoretical foundation to support the proposed simple method works well.

**Questions:**

Please address the weakness above.

**Details Of Ethics Concerns:**

Code comments written in Chinese is not that great for anonymity.